# Experimental Study of the Influence of Different Load Changes in Inlet Gas and Solvent Flow Rate on CO_2_ Absorption in a Sieve Tray Column

**DOI:** 10.3390/e24091318

**Published:** 2022-09-19

**Authors:** Adel Almoslh, Babak Aghel, Falah Alobaid, Christian Heinze, Bernd Epple

**Affiliations:** Institut Energiesysteme und Energietechnik, Technische Universität Darmstadt, Otto-Berndt-Straße 2, 64287 Darmstadt, Germany

**Keywords:** CO_2_ absorption, tray pressure drop, liquid holdup level, liquid holdup, experimental study, entropy generation

## Abstract

An experimental study was conducted in a sieve tray column. This study used a simulated flue gas consisting of 30% CO_2_ and 70%. A 10% mass fraction of methyl diethanolamine (MDEA) aqueous solution was used as a solvent. Three ramp-up tests were performed to investigate the effect of different load changes in inlet gas and solvent flow rate on CO_2_ absorption. The rate of change in gas flow rate was 0.1 Nm^3^/h/s, and the rate of change in MDEA aqueous solution was about 0.7 NL/h/s. It was found that different load changes in inlet gas and solvent flow rate significantly affect the CO_2_ volume fraction at the outlet during the transient state. The CO_2_ volume fraction reaches a peak value during the transient state. The effect of different load changes in inlet gas and solvent flow rate on the hydrodynamic properties of the sieve tray were also investigated. The authors studied the correlation between the performance of the absorber column for CO_2_ capture during the transient state and the hydrodynamic properties of the sieve tray. In addition, this paper presents an experimental investigation of the bubble-liquid interaction as a contributor to entropy generation on a sieve tray in the absorption column used for CO_2_ absorption during the transient state of different load changes.

## 1. Introduction

The absorption technology for CO_2_ capture mainly consists of the absorber column and the regeneration unit. The absorber column can be a packed column or a plate column. The solvent enters the absorber from the top, and the waste gas containing CO_2_ enters the absorber from the bottom. The gas and liquid phases contact each other on the packing materials or the plates. The packing material or the plates increase the gas-liquid interface, which improves mass and heat transfer between the contact phases. The CO_2_ component flows from the gas phase to a liquid phase and is then absorbed. There are multiple parameters that influence the absorption process, e.g., pressure, temperature, gas flow, and solvent flow. In a steady state, these parameters are typically fixed, but in a transient behavior, these parameters are changed, e.g., different load changes in inlet gas and solvent flow rate start-up or shutdown of the plant. It is vital to determine the behavior of the process during this dynamic change to improve the control circuits’ performance or to identify the problems that may occur during the transient state.

Solvents with various additives can be used for CO_2_ absorption. Amine solvents such as monoethanolamine (MEA), diethanolamine (DEA), and methyl diethanolamine (MDEA) are widely used for CO_2_ capture [1]. Skylogianni and Eirini et al. investigated the solubility of carbon dioxide in non-aqueous and aqueous mixtures of methyl diethanolamine (MDEA) with mono ethylene glycol (MEG). The authors concluded that the absorption capacity of the aqueous solvents at a constant amine concentration declined with increasing glycol concentration and with the substitution of water, increasing the amine concentration up to 90 wt% in aqueous MDEA systems, also resulting in less CO_2_ loading in the solvents. The authors stated that in the non-aqueous solvents, a transient phase was noticed for compositions between 30 and 50 wt% MDEA-MEG; CO_2_ solubility enhances with amine concentration up to this range, after which solubility begins to decrease. The authors related this behavior to CO_2_ capture by the chemical reaction of CO_2_ with MEG in the presence of MDEA as a result of the auto-protolysis of MEG in the alkaline environment of the amine [2].

In the literature, various studies can be found on the influence of the inlet gas flow on the hydrodynamic characteristics and mass transfer in gas-liquid systems. Some studies discuss the influence of gas velocity on froth height and the height of clear liquid on the trays.

Dhulesia, H. (1984) [3] tested the effect of gas velocity on clear liquid height for three sieve trays. They plotted the height of the clear liquid versus the flow ratio Ψ^0.25, and they found that the height of the clear liquid was proportional to the flow ratio for the froth regime. Nevertheless, for the spray regime, they stated that the dependence of clear liquid height on the flow ratio group could not be detected. Dhulesia, H. (1984) [3] also investigated the influence of liquid and gas rates on the clear liquid height by using a weir height of 25 mm valve tray. They established that the clear liquid height increases with liquid volume, whereas the clear liquid height decreases with increasing superficial air velocity.

Badssi, Bugarel et al., 1988 [4] explored the effect of the superficial velocity of gas and liquid on the interfacial area in two gas-liquid systems: CO_2_-DEA and CO_2_-NaOH. The experiment was carried out in a laboratory column equipped with crossflow sieve trays. They found that the total interfacial area increased when the superficial velocity was increased.

Wijn (1999) [5] stated that the liquid height depends on the gas and liquid loads, gas and liquid properties, and some geometrical parameters such as the height and length of the weir, free hole area, hole diameter, etc.

Van Baten, Ellenberger et al., (2001) [6] investigated the hydrodynamics of a sieve tray column for reactive distillation. The author observed that the clear liquid decreased significantly when the superficial velocity of the gas increased between 0.4 and 1 m/s.

Furzer (2001) [7] determined the height of froth and the height of clear liquid on dual-flow trays with 20% free area; the authors stated that there is a strong relationship between the height of froth and the height of the clear liquid, as the height of froth increases when the height of clear liquid increases. The authors noted that the height of the clear liquid increases with the vapor velocity.

Zarei et al., 2010 [8] studied hydraulic parameters such as dry pressure drop in a column with a diameter of 1.22 m. The column has two sieve trays and two chimney trays; the author observed that the pressure drop increases when the Fs factor is increased. Their experiments were conducted in a round tower with a diameter of 1.22 m for the air/water system; the author observed that the clear liquid’s height decreases as the gas velocity increases.

R. Brahem (2015) [9] reviewed the experimental measurements of hydrodynamic and interface parameters performed on two pilot-scale rectangular valve tray columns. They present their results for the height of the clear liquid as a function of the flow ratio Ψ and show that the height of the clear liquid increases as the flow ratio Ψ increases. In the same study, they also plot that the tray pressure drop increases by increasing the gas kinetic factor Fa.

Kurella, Bhukya et al., 2017 [10] studied the effect of the gas velocity on the average height of the clear liquid on the tray; their experimental study was conducted in a dual-flow sieve plate scrubber. The authors found that at constant liquid flow rates, the average clear liquid height increased as the gas loading factor (Fs) was increased. Kurella, Bhukya et al., 2017 [10] also examined the effects of gas and liquid flow rates on the percent removal of H2S at H2S input concentrations of 50–300 ppm. Their experiments were performed in a lab-scale three-stage dual-flow sieve plate column scrubber. The authors concluded that the percentage of H2S removal increases as the gas flow rate increases.

Feng, Fan et al., 2018 [11] analyzed the effects of the Fs factor on dry pressure drop, wet pressure drop, clear liquid height, and froth height. Their experiments were conducted using a folding sieve tray (FST), which consists of double-perforated oblique planes folding at a specific angle. The author found that the dry pressure drop, wet tray pressure drop, clear liquid height, and froth height increased when the Fs factor of the gas was increased, while the clear liquid height decreased when the Fs factor was increased.

Almoslh, Alobaid et al., 2021 [12] investigated the effect of gas flow rate on the hydrodynamic characteristics of the sieve tray. The authors investigated the hydrodynamic characteristics of the sieve tray for the gas/water system at different gas flow rates from 12 to 24 Nm^3^/h and at different pressures of 0.22, 0.24, and 0.26 MPa. The authors observed that liquid holdup increased with an increase in gas inlet flow rate between 12 and 20 Nm^3^/h, while an increase in gas inlet flow rate between 20 and 24 Nm^3^/h generated no increase in the liquid holdup. The authors studied the effect of changing these hydrodynamic characteristics on the performance of a tray column used for CO_2_ capture. They found that increasing liquid holdup by increasing the gas inlet flow rate improved the performance of the CO_2_ absorber.

Different sources are responsible for entropy generation, such as heat, mass, and momentum transfer on the trays. Energy is also lost through the resistance of the liquid on the inner sides of the tray internals [13].

The paper by Chermiti et al., 2014 [14] deals with the analysis of entropy generation in the case of chemical absorption of a gas in a laminar falling liquid film. CO_2_ absorption in aqueous solutions of monoethanolamine (MEA) was investigated. Their obtained results reveal that the major contributor to entropy generation is chemical reaction irreversibility at the gas-liquid interface.

S. RAY et al., 1996 [13] reviewed a theoretical analysis of the separation process on a single sieve tray. The bubble-liquid interaction on the tray is the main contributor to irreversibility on a sieve tray in a distillation column for the separation of light hydrocarbons. In the case of the bubbles that form at the holes of the sieve bottom, most of the entropy generation takes place before the bubbles detach.

The steady-state design of chemical plants encounters problems of dynamics and controllability issues. To avoid wrong assumptions in process synthesis and design and to ensure safe start-up and shutdown as well as stable plant operation, the dynamic behavior of the units involved should be known [15]. The purpose of studying the dynamic state is to monitor the model’s predictive ability in the presence of various changes and disturbances. Transient conditions include changes in production rate, rapid changes in temperature or pressure, and also variations in composition; this is important for process control design to ensure the optimization of process equipment [16].

Harun et al., 2012 [17] developed a dynamic MEA absorption process model to investigate the dynamic behavior of the CO_2_ capture process. Harun et al., 2012 [17] studied the behavioral response of the monoethanolamine (MEA) absorption process during the transient state of changes in flue gas flow rate and reboiler heating power. The authors found that the changes in flue gas flow rate and reboiler heating power are major process parameters that affect the percentage of CO_2_ removal, liquid-to-gas ratio, and lean loading. Their results reveal that the variation between the reboiler heating capacity and CO_2_ removal is about 1:1.

Lawal et al., 2009 [18] developed and compared two models (equilibrium-based and rate-based models). Their study is conducted on post-combustion CO_2_ capture using monoethanolamine (MEA) as a solvent. This study aims to understand the absorber’s dynamic behavior during partial load operation and when the stripper is disturbed. Lawal reviewed that the rate-based model is more accurate in prediction than the equilibrium-based model. Lawal found that absorber operation is more responsive to the L/G ratio. The authors reviewed that increased CO_2_ loading in the lean solvent resulted in a significant reduction in absorber performance.

Gaspar and Cormos (2012) [19] developed a rate-based model for simulating the CO_2_ post-combustion process using amine-based solutions in a fixed-bed absorption column. The objective of this study was to investigate the dynamic behavior and absorption performance of four different types of alkanolamines (MEA, DEA, MDEA, and AMP), using mass transfer and liquid holding correlation models published in the literature, such as the model by Wang et al., the Billet and Schultes model, and the Rocha model. The authors found that the mass transfer correlation model proposed by Wang et al. well predicted the effective mass transfer area and the mass transfer coefficient correlation for all alkanolamines.

Gáspár and Cormoş 2011 [20] performed modeling and simulation of the CO_2_ absorption and regeneration process using abundant amine. This study aims to validate models and understand the dynamic behavior of the whole capture and regeneration stages. One of the cases studied by the authors is the change in power plant load by linearly increasing the ratio of gas flow rate to liquid flow rate (FG/FL) from 625 to 1040. The authors found that the amount of purified CO_2_ gas increases with the power plant load, but the exhaust gas stream is richer in CO_2_. The authors reviewed that the developed model could predict the dynamic behavior of the columns during operation.

To the author’s knowledge, there are only a limited number of studies in the literature dealing with the dynamic state of the absorption process, and these need to be enriched. Therefore, the objective of this study is to experimentally investigate the behavior of the absorber used for CO_2_ absorption during the transient state under different load changes in inlet gas and solvent flow rate. An absorber test stand was built and operated to experimentally investigate the influence of different load changes on CO_2_ absorption. The derivative objectives of this study are to investigate the influence of different load changes in inlet gas and solvent flow rate on the CO_2_ volume fraction at the outlet and also to investigate the effect of different load changes in inlet gas and solvent flow rate on the hydrodynamic properties of the sieve tray, such as total tray pressure drop, liquid holdup level and liquid holdup. In addition, one of the objectives of this experimental study is to study the relationship between the hydrodynamic characteristics of a sieve tray and the performance of a tray absorber for CO_2_ capture under different load changes in inlet gas and solvent flow rate.

## 2. Experimental Section

### 2.1. Test Rig Setup

Figure 1 and Figure 2 show an absorber test rig that was constructed at Technische Universität Darmstadt. The absorber test rig consists of four main parts: an absorber column, a gas analysis unit, a gas mixing unit, and a regeneration unit. The absorber is made of a glass column that has an inner diameter of 152 mm and a height of 1500 mm. The top and the bottom of the column were clogged by appropriate metal flanges. The bottom flange contains the exit of liquid, and the top flange contains the exit of the gas. The column holds 12 glass nozzles to which metal flanges can be combined, ten nozzles of them are employed to measure pressures and temperatures in the absorber, and two nozzles for entering the inlet gas and liquid to the absorber. Five sieve trays are attached by threaded rods and joined inside the absorber. Figure 3 illustrates the geometry of the tray used in this study. The diameter of the tray is 150 mm, the distance between the tray and glass wall is stuffed with rubber seals. The bubble area on the tray is 0.013 m^2^. The diameter of the holes is 2 mm. The height of the weir is 15 mm. The distance between the trays is 240 mm. The mixing unit consists of two tubes connected to a manifold upstream of the absorber. One of the tubes coupled with cylinders is supplied with CO_2_ gas, and the other is attached to an air compressor. Mass flow controllers are employed to control the volumetric flow rate of gases entering the absorber. A gas analysis unit is bound at the gas outlet tube to measure the volumetric fraction of CO_2_ at the outlet of the absorber. The gas analysis unit is manufactured by M&C TechGroup from Ratingen, Germany.

The solvent regeneration unit aims to regenerate the solvent and recycle it to the absorber as a lean solvent. It consists of two heat exchangers, a packed column, a reboiler, a makeup pump, and a recycle pump. The packed column was constructed of a glass column with a height of 1300 mm, and a diameter of 152 mm. The packed column is filled with a metal packing material from type Pall-Ring 15 mm with a specific surface of 360 m^2^/m^3^, and a free volume of 95 [%]. The height of the packing material is 1 m. The rich solvent enters the packed column at the top within a liquid distributor. The objective of the liquid distributor is to spread the solvent uniformly on the top of the packing material; the created liquid distributor is a spray with 13 holes collected uniformly on the liquid distributor. The packed column is seated on the reboiler. The reboiler has a cylindrical shape, a diameter of 220 mm, and a length 606 mm. The reboiler was manufactured at the Institute of Energy Systems and Energy Technology workshop, Technical University of Darmstadt. The reboiler is made of stainless steel because it is corrosion-resistant to amine solvents. The volume of the reboiler is about 24 L. A coil with thermal power of 4.5 kW was inserted into the reboiler to regenerate the solvent. The circulating pump is of the LCA1 type, its manufacturer is LEWA GmbH from Leonberg, Germany. The circulating pump is attached to the reboiler, which draws the solvent from the reboiler and pumps it to the absorber. The lean hot solvent is cooled down by two heat exchangers, by the first heat exchanger, the lean hot solvent is precooled by exchanging heat with the solvent, which escapes from the absorber, whereas in a second heat exchanger, the precooled solvent cools down by exchanging heat with cool water.

### 2.2. Instrumentation and Control Equipment of the Test Rig

The test rig is supplied with various instruments and control circuits, which are seated to gauge the needed parameters of the absorption process and for secure operation. A pressure reducer is set on every line of the gas mixer to modify the maximum pressure of gas coming in the absorber. Behind a pressure reducer, a magnetic valve is set, to allow the opening or closing of the gas supply which can be closed at emergency conditions. An MFC is fixed on every line of the gas mixer to modify the volumetric flow rate. To measure the temperature of the fluid on the tray, a temperature sensor is set near each absorber tray. A Coriolis device is fixed at the inlet of the liquid to measure the temperature and solvent flow rate entering the absorber. To measure the total tray pressure drop, a pressure difference device was fixed at the third tray of the absorber column.

The test rig is fitted with five control circuits. The first control circuit is to maintain the pressure to the setpoint. The pressure control circuit consists of a control valve and a pressure sensor. The control valve is positioned at the absorber’s gas outlet, whereas the pressure sensor is placed at the column. The pressure sensor is located exactly between the first and second tray and is numbered from top to bottom of the absorber. The pressure control circuit controls the pressure after the gas enters the absorber. The pressure sensor sends a signal with the actual pressure value to a PID controller. The PID controller compares the set point of pressure and the actual value of pressure and signals the pneumatic control valve to open or close, maintaining the pressure at its set point. For safety reasons, a safety pressure valve is installed in the gas outlet of the absorber column. The manufacturer calibrated the safety pressure valve to a value of 0.45 MPa. Thus, when the pressure reaches a value of 4.5 and above, the safety pressure valve opens to release the gas in the absorption column, helping to reduce the pressure to a value lower than 0.45 MPa.

The second control circuit has the task of maintaining the liquid level at the bottom of the absorber. Maintaining the liquid level is necessary because otherwise the liquid will accumulate in the absorber and interrupt the gas flow in the absorber. The level control circuit consists of a pressure differential device and a control valve. The pressure differential device is located in the sump of the column, while the control valve is located at the liquid outlet of the absorber. The third control circuit controls the level of the solvent in the reboiler since some loss of solvents occurs due to the evaporation of water. The third control loop consists of a level sensor and a make-up pump. The level sensor sends a signal to the make-up pump when the solvent level falls below the set point; in this case, the make-up pump pumps fresh solvent into the reboiler. The fourth control circuit is provided to control the temperature of the solvent in the reboiler. The purpose of this control circuit is to regenerate the solvent by heating it with the help of a heater installed in the reboiler. For unexpected reasons, the fourth control circuit may not work; in order to save the heater, a fifth control circuit is installed in the reboiler, which switches off the heater when the liquid level in the reboiler falls below the solvent level set point.

### 2.3. Test Procedure

Since the objective of the study is to investigate the effect of different load changes in inlet gas on the acid gas removal system, CO_2_ was selected as one of the acid gases. In the industry, carbon dioxide gas often found in a mixture of gases with different concentrations and acid gases are selectively removed using solvents such as amine solvents. Therefore, it would be ideal for the study if different gases were mixed and introduced into the absorber. However, since the instrument can currently only mix two gases, the rest of the gases that make up the gas mixture were replaced with nitrogen gas, and the reason for choosing hydrogen gas is that it is an inert gas that does not interact with the solvent.

In the experiment, the CO_2_ gas was mixed with N_2_ gas using the gas mixing unit before entering the absorber column. The N_2_ is an inert gas and serves as a carrier gas. The CO_2_ volume fraction was 0.3 in all experiments. MDEA aqueous solution with a 10% mass fraction was used as a solvent. The input gas flow rate was constant at 14 Nm^3^/h at the start. The pressure of the absorber was constant at 0.28 MPa. The inlet solvent flow rate was constant at 94 NL/h at the start. The temperature of the inlet solvent was controlled at about 20 °C. The regeneration unit was operated with a heating power of 4.5 kW during all experiments.

Figure 4 shows three ramp-up tests which were performed to investigate the effect of different load changes in inlet gas and solvent flow rate on CO_2_ absorption. The ramp-up tests were performed by changing the operation of the absorber from a stationary state (1) to a stationary state, and (2) to a transient state. During the transient state, both the gas and solvent flow rate gradually increased between the stationary state (1) and stationary state (2) with a certain rate of change in the flow rate. In the transient state of the ramp-up tests, the gas flow rate was increased from 14 Nm^3^/h to 20 Nm^3^/h with a rate of change of about 0.1 Nm^3^/h/s, where the median flow rate of the incoming solvent were increased from 94 NL/h to 114 NL/h for ramp-up test (1) and to 131 NL/h, and 150 NL/h for ramp-up test (2) and (3) sequentially. The rate of change in the solvent flow rate is about 0.7 NL/h/s.

The absorber test rig is operated under specified conditions for 20 min during each measurement, resulting in time-dependent values for each measured parameter (i.e., gas concentrations, temperature, and pressure). The standard deviation, which gives the range of deviation of each measured parameter, is then estimated to evaluate the random error. The systematic error of the measuring devices is constant for all tests and is therefore not presented further in this chapter. In general, the uncertainty of directly measured values (e.g., gas concentrations, temperature, and pressure) depends only on the relative uncertainty of the measuring instruments and is expressed by the relative error. For indirectly measured parameters or calculated values (e.g., volumetric flow, where the pressure difference and temperature are taken into account in the calculation), the Gaussian error propagation method is used, assuming normally distributed uncertainties. In this study, the volumetric concentrations are measured by the gas analyzer, and the maximum relative error for CO_2_ in the different process streams is about 3%.

## 3. Results and Discussion

### 3.1. Effect of Different Load Changes in Inlet Gas and Solvent Flow Rate on Outlet CO_2_ Volume Fraction

Figure 5 shows the effect of different load changes in inlet gas and solvent flow rate on the outlet volume fraction of CO_2_. It is shown that different load changes significantly affect the volume fraction of CO_2_. It is noted that the CO_2_ volume fractions before the transient state were close to 0.248 in all start-up tests. During the transient state, CO_2_ volume fractions increased rapidly and reached a peak value in all ramp-up tests. The height of the peak decreases as different load changes are increased. After the transient state, the CO_2_ volume fraction stabilizes at 0.255 for the ramp-up test (1) and 0.253 and 0.251 for the ramp-up test (2) and (3), respectively.

The explanation for this behavior of the absorber is that before the transient state, the operation conditions were similar in all ramp-up tests, so the mole fraction of CO_2_ is similar in all tests. During the transient state, the dynamic change in gas flow rate leads to a faster increase in liquid holdup level and liquid holdup in the upper trays than in the lower trays. As a result, the lower trays have lower efficiency than the upper trays, leading to a decrease in absorber performance, and the CO_2_ mole fractions reach a peak during the transient state. To verify this explanation, one needs to investigate the effect of different load changes in inlet gas and solvent flow rate on the tray’s hydrodynamic properties, such as tray pressure drop, tray holdup level, and liquid holdup.

### 3.2. Effect of Different Load Changes in Inlet Gas and Solvent Flow Rate on Hydrodynamic Characteristics of the Tray

#### 3.2.1. Effect of Different Load Changes in Gas and Solvent Flow Rate on Total Pressure Drop

To study the effect of different load changes in inlet gas and solvent flow rate on the hydrodynamic characteristics of the tray, the absorber test rig, as seen in Figure 2 is equipped with a pressure difference transmitter that measures the total pressure drop at the third tray. The total pressure drop is the sum of the dry tray pressure drop and the hydraulic tray pressure drop is as follows:(1)$$\Delta P_{\mathrm{total},\mathrm{tray}}=\Delta P_{\mathrm{dry},\mathrm{tray}}+\Delta P_{\mathrm{hyd}.\mathrm{tray}}$$
where $\Delta P_{\mathrm{total},\mathrm{tray}}$ is the total tray pressure drop, $\Delta P_{\mathrm{dry},\mathrm{tray}}$ is the dry tray pressure drop, and $\Delta P_{\mathrm{hyd}.\mathrm{tray}}$ is the hydraulic tray pressure drop.

$\Delta P_{\mathrm{total},\mathrm{tray}}$ is measured during performing the experiments when the solvent and the gas enter the column, whereas $\Delta P_{\mathrm{dry},\mathrm{tray}}$ is measured when only the gas enters the column at the same other operating conditions from pressure and inlet solvent flow rate.

Figure 6 shows the effect of different load changes of the gas and solvent flow rate on the total tray pressure drop. It can be seen that the total tray pressure drop increases rapidly during the transient state from 0.55 kPa to 0.65 kPa for the ramp-up test (1) and 0.67 kPa and 0.7 kPa for the ramp-up test (2) and (3), respectively. The explanation for this behavior is that an increase in the gas flow rate leads to an increase in the superficial velocity of the gas. The increase in the superficial velocity of the gas will improve trapping the liquid on a tray and causes the liquid to accumulate on the tray, which leads to an increase in the total tray pressure drop.

#### 3.2.2. Effect of Different Load Changes in the Inlet Gas and Solvent Flow Rate on the Tray Holdup Level and Tray Liquid Holdup

The liquid holdup is the amount of the liquid trapped or accumulated on the tray during the operation of the absorber. In order to estimate the tray holdup level and tray liquid holdup, the measured total pressure drop in the tray is required. From Equation (1), the $\Delta P_{\mathrm{hyd}.\mathrm{tray}}$ can be obtained as follows:(2)$$\Delta P_{\mathrm{hyd}.\mathrm{tray}}=\Delta P_{\mathrm{total},\mathrm{tray}}-\Delta P_{\mathrm{dry},\mathrm{tray}}$$

$\Delta P_{\mathrm{total},\mathrm{tray}}$ is measured during performance of the ramp-up tests when the solvent and the gas enter the column, where $\Delta P_{\mathrm{dry},\mathrm{tray}}$ is measured when only the gas enters the column at operating conditions of gas flow rates between 12 Nm^3^/h and 20 Nm^3^/h and at a pressure of 0.28 MPa. The measurements of dry tray pressure drop are shown in Table 1.

Since the hydraulic tray pressure drop is equivalent to the hydrostatic height of the liquid on the tray, the hydrostatic height of the liquid on the tray can be calculated as follows:(3)$$h_{\mathrm{cl}}=\Delta P_{\mathrm{hyd}.\mathrm{tray}}\times 10.2$$
where $h_{\mathrm{cl}}$ is liquid holdup level or the hydrostatic height of the liquid on the tray (m), $\Delta P_{\mathrm{hyd}.\mathrm{tray}}$ is the hydraulic tray pressure drop (kPa) and 10.2 is a constant for converting the unit of kPa to (cm).

From Equation (3), liquid holdup can be calculated as follows:(4)$$h_{L}=A_{\mathrm{tray}}\times h_{\mathrm{cl}}\times 0.001$$

$h_{L}$ is liquid holdup (Liter) and $A_{\mathrm{tray}}$ is the tray area (cm^2^), 0.001 is a constant for converting the unit of cm^3^ to (Liter).

Figure 7 and Figure 8 show the effect of different load changes in the gas and solvent flow rate on tray liquid holdup level and tray liquid holdup during the transient state. It can be seen that both tray holdup level and tray liquid holdup increase rapidly during the transient state. From Figure 7, the liquid holdup level increases during the transient state from 5.7 cm to 6 cm for the ramp-up test (1), and to 6.4 cm and 6.8 cm for the ramp-up test (2) and (3), respectively. From Figure 8, the liquid holdup increases during transient state from 1 L to 1.1 L for the ramp-up test (1), and to 1.2 L and 1.25 L for the ramp-up test (2) and (3), respectively. The explanation for this behavior is that the increase in the flow rate of the gas leads to an increase in the superficial velocity of the gas; the increase in the superficial velocity of the gas will improve trapping the liquid on a tray and causes the liquid to accumulate on the tray and increases the holdup level tray and liquid holdup. The effect of different load changes in the solvent flow rate are not apparent during the transient state, since the estimated tray holdup level and tray liquid holdup are applied to the third tray. The solvent flow across the absorber is slower than that of the gas, so the effect of different load changes in the solvent takes some time to appear on the third tray.

It can also be noted that the settling time of the liquid holdup level tray and the liquid holdup on the third tray is longer than the time of the transient state of different load changes in the inlet gas and solvent flow rate. Although the transient state of the gas and solvent flow continued for 40–60 s, the hydrodynamic properties continue to change dynamically for about 10–20 s after the transient state of gas and solvent flow. It is assumed that the delay in settling time for the liquid holdup level tray and the liquid holdup influences the absorber’s performance during the transient state. In order to verify this assumption, it is necessary to study the absorber’s performance during the transient state, which will be studied later.

#### 3.2.3. Entropy Generation Due to Bubble Bursting

The gas bubble grows when the gas stream passes through the sieve holes because the gas pressure is higher than the hydrostatic pressure of the tray. The diameter of the holes and their number and distribution on the tray surface play an important role in determining the bubble’s diameter and the number of bubbles that form after the gas flow penetrates the tray. After bubble formation, the bubble moves vertically to the surface of the liquid on the tray. The gas pressure and gas flow rate determine the velocity at which the bubble rises through the liquid. As the flow rate of the gas entering the absorption column increases, the velocity of the gas increases, resulting in an increase in the velocity of the bubble movement through the liquid. The diameter of the bubble increases as it rises toward the surface of the liquid on the tray as the hydrostatic pressure decreases toward the surface of the liquid.

Increasing the bubble’s diameter as it moves vertically increases the possibility that the bubbles will touch and merge to form larger diameter bubbles. The acceleration of the bubble increases as it approaches the surface of the liquid. As the bubble approaches the surface, it bursts. The vertical motion of the bubble and its bursting at the surface transfers momentum to the tray’s surrounding liquid. S. RAY et al., 1996 [13] review the entropy generation when bubbles burst as follows:(5)$$sg=\sigma \cdot A_{b}/\left(T_{1}\cdot m_{bf}\right)$$
where *sg* is entropy generation due to bubble bursting, *σ* is the surface tension of the liquid [J m^−2^], *A_b_* is the surface area of the bubble [m^2^], *T*_1_ temperature (absolute) of liquid, bubble and vapor upstream of the sieve tray [K], *m_bf_* is the final mass of the bubble [kg].

From this correlation, it can be concluded that the entropy generation increases with the increase in the bubble surface area. In order to investigate the effect of different load changes in the gas flow rate on the entropy generation, the froth height was studied, mainly generated by the bursting of the bubbles on the tray during the transition state. For this purpose, the absorber was fitted with a ruler to observe the froth formation above the tray. From Figure 9, it was observed during the test that the froth height increased during the ramp-up test. It can also be seen that the froth height above the tray increased during the ramp-up test.

This behavior can be explained as follows: The velocity of the gas in the absorber increases due to the increase in the gas flow rate, which increases the number of growing bubbles as well as the bursting of the bubbles, which leads to entropy generation. The increase in foam height can be seen as an indicator that the interface between the gas and liquid phases is increasing, leading to an increase in mass transfer and the amount of CO_2_ absorbed.

### 3.3. Effect of Different Load Changes in the Inlet Gas and Solvent Flow Rate on the Performance of the Absorber

The performance of the absorber for CO_2_ capture was evaluated by estimating the absorption percentage of CO_2_. The absorption percentage of CO_2_ was calculated using the equation as follows:(6)$$N_{{\mathrm{CO}}_{2}}=F_{\mathrm{gas},\mathrm{in}}y_{\mathrm{co}2,\mathrm{in}}-F_{\mathrm{gas},\mathrm{out}}y_{\mathrm{co}2,\mathrm{out}}$$$F_{\mathrm{gas},\mathrm{out}}$ can be estimated as follows:(7)$$F_{\mathrm{gas},\mathrm{out}}=F_{\mathrm{gas},\mathrm{in}}-N_{{\mathrm{CO}}_{2}}$$Substituting Equation (7) into Equation (6), one obtains:(8)$$N_{{\mathrm{CO}}_{2}}=F_{\mathrm{gas},\mathrm{in}}y_{\mathrm{co}2,\mathrm{in}}-\left(F_{\mathrm{gas},\mathrm{in}}-N_{{\mathrm{CO}}_{2}}\right)y_{\mathrm{co}2,\mathrm{out}}$$Or
(9)$$N_{{\mathrm{CO}}_{2}}=F_{\mathrm{gas},\mathrm{in}}\left(y_{\mathrm{co}2,\mathrm{in}}-y_{\mathrm{co}2,\mathrm{out}}\right)/\left(1-y_{\mathrm{co}2,\mathrm{out}}\right)$$
(10)$${\mathrm{Absorption}\mathrm{percentage}\mathrm{of}\mathrm{CO}}_{2}=N_{{\mathrm{CO}}_{2}}x100/F_{{\mathrm{CO}}_{2},\mathrm{in}}$$
where $N_{{\mathrm{CO}}_{2}}$ is the absorbed rate of CO_2_
$\left[{\mathrm{Nm}}^{3}/h\right]$, $y_{{\mathrm{CO}}_{2},\mathrm{in}}$ is the inlet volumetric fraction of CO_2_, $y_{{\mathrm{CO}}_{2},\mathrm{out}}$ is the outlet volumetric fraction of CO_2,_
$F_{\mathrm{gas},\mathrm{out}}$ is the outlet gas flow rate $\left[{\mathrm{Nm}}^{3}/h\right]$, $F_{\mathrm{gas},\mathrm{in}}$ is the inlet gas flow rate $\left[{\mathrm{Nm}}^{3}/h\right]$, and $F_{{\mathrm{CO}}_{2},\mathrm{in}}$ is the inlet CO_2_ flow rate $\left[{\mathrm{Nm}}^{3}/h\right]$.

The $y_{{\mathrm{CO}}_{2},\mathrm{out}}$ was measured by the gas analysis unit, where $y_{{\mathrm{CO}}_{2},\mathrm{in}}$ was calculated as follows:(11)$$y_{{\mathrm{CO}}_{2},\mathrm{in}}=\frac{F_{{\mathrm{CO}}_{2},\mathrm{in}}}{F_{\mathrm{gas},\mathrm{in}}}=\frac{F_{{\mathrm{CO}}_{2},\mathrm{in}}}{F_{{\mathrm{CO}}_{2},\mathrm{in}}+F_{\mathrm{air},\mathrm{in}}}$$
where $F_{\mathrm{air},\mathrm{in}}$ is the inlet air flow rate $\left[{\mathrm{Nm}}^{3}/h\right]$.

Figure 10 illustrates the effects of different load changes in the inlet gas and solvent flow rate on the absorption percentage of CO_2_. It can be seen that different load changes have a significant effect on CO_2_ absorption. During the transient state, the absorption percentage of CO_2_ decreased rapidly.

To evaluate the absorber’s performance during the transient state, it can be observed that although the solvent flow rate was increased during the transient state, the CO_2_ volume fractions reached a peak value in all ramp-up tests. The main reason for this behavior is the delayed response of the absorber to the absorption of excess CO_2_ during the transient state. This delay appears by the settling time of hydrodynamic characteristics of the tray, as seen in Figure 7 and Figure 8. It can be seen that the settling time of the hydrodynamic characteristics of the tray is longer than the time of the transient state of different load changes in the inlet gas and solvent flow rate. Thus, it can be concluded that there is a significant relationship between the trays’ hydrodynamic properties and the absorber’s performance. During the transition state, the liquid level in the upper trays increases faster than in the lower trays, so the lower trays contain less liquid during the transient state and thus have a lower efficiency.

It can also be concluded that the absorber’s performance during the transient state of the ramp tests is not optimal. One of the proposed solutions to improve the absorber’s performance during the transient state is to start the different load changes in the solvent flow rate before a certain period of starting different load changes in the gas flow rate. This period can be determined based on the settling time of the hydrodynamic properties of the tray. In this case, the trays will fill with the required amount of liquid holdup before starting the different load changes in the gas flow rate. In this way, it is assumed that the absorber will absorb the excess CO_2_ during the transient state and the CO_2_ volume fractions may not reach a peak value.

## 4. Conclusions

An absorber test rig was constructed and operated. The effect of different load changes in the inlet gas and solvent flow rate on the absorber’s performance to capture CO_2_ was experimentally studied. In addition, the effect of the hydrodynamic properties of a sieve tray on CO_2_ absorption in the transient state was investigated, highlighting the following points:

The bubble-liquid interaction contributes to irreversibility on a sieve tray in the absorption column used for CO_2_ absorption during the transient state of different load changes.

(1)Most of the entropy generation is due to the formation of bubbles and their vertical movement through the liquid on the sieve tray, and bursting near the liquid surface.(2)The CO_2_ volume fractions peak during the transient state of different load changes in the inlet gas and solvent flow rate.(3)The hydrodynamic characteristics of the tray increase during the transient state. The settling time of the hydrodynamic characteristics is longer than the time of the transient state of different load changes in the inlet gas and solvent, which decrease the absorber’s performance.(4)During the transient state, the absorption percentage of CO_2_ decreased. It can be determined that the absorber’s performance is not optimal during the transient state of the ramp tests.(5)There is a significant correlation between the hydrodynamic characteristics of a sieve tray and the absorber’s performance during the transient state. The liquid level in the upper stage increases faster than in the lower stage during the transient state, so the lower trays have less liquid and, thus, lower efficiency during the transient state. After the transient state, the liquid level will stabilize at a certain level. The delay in settling time for the liquid holdup level tray and the liquid holdup has an unfavorable influence on the absorber’s performance during the transient state.

## Figures and Tables

**Figure 1 entropy-24-01318-f001:**
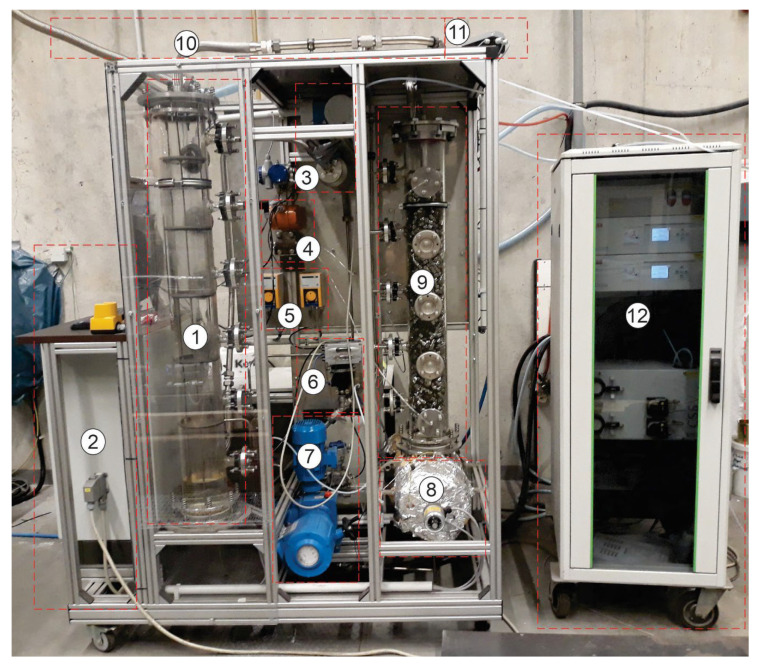
Side view of the absorber test rig: 1, absorber column; 2, control panel; 3, Coriolis device; 4, pressure difference transmitter; 5, make-up pump; 6, liquid level control valve; 7, recycling pump; 8, re-boiler; 9, packed column; 10, gas outlet; 11, pressure control valve; 12, gas analysis unit.

**Figure 2 entropy-24-01318-f002:**
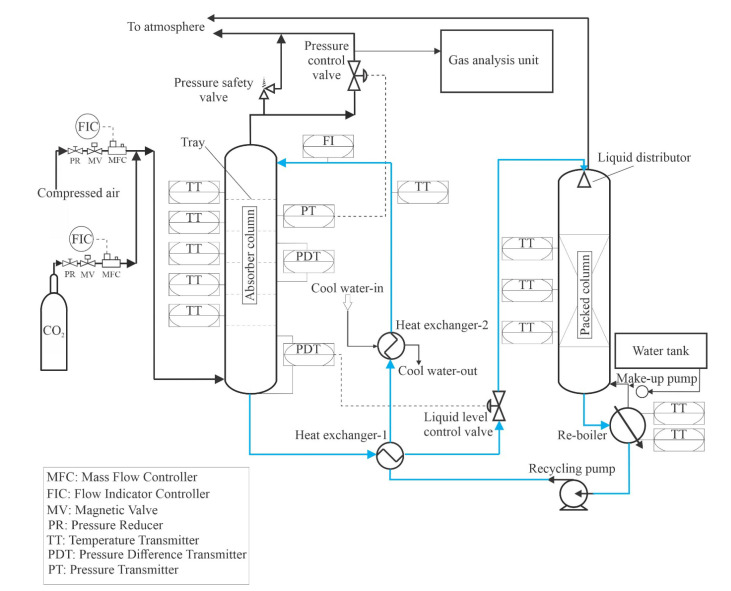
Schematic diagram of the absorber test rig.

**Figure 3 entropy-24-01318-f003:**
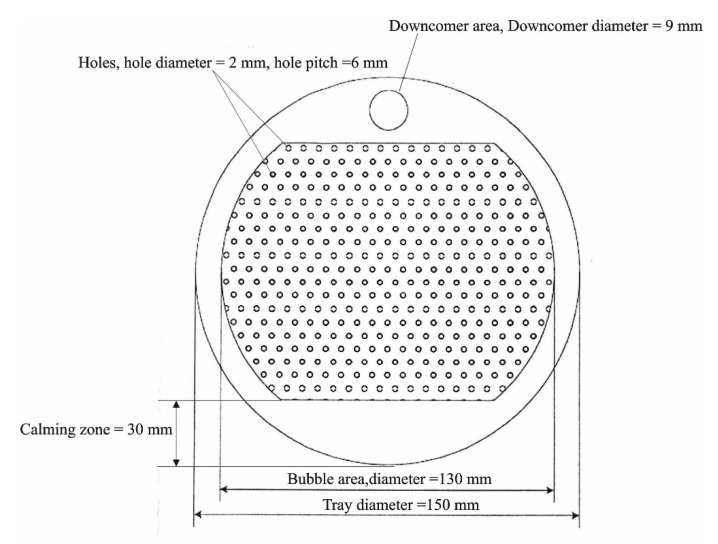
Illustration of the geometry of the tray.

**Figure 4 entropy-24-01318-f004:**
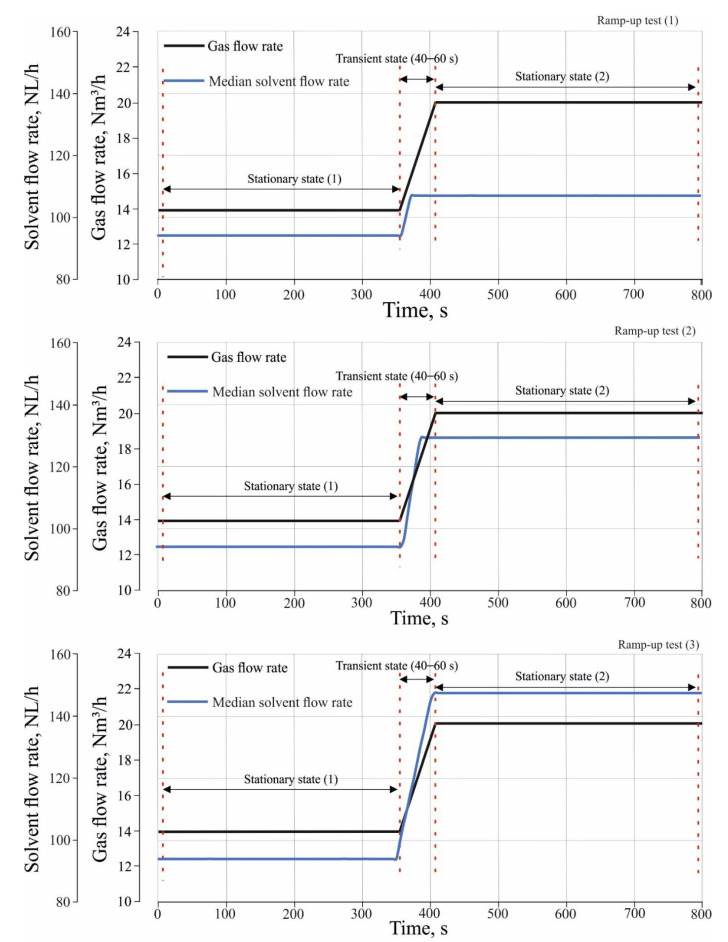
The ramp-up tests of different load changes in inlet gas and solvent flow rate.

**Figure 5 entropy-24-01318-f005:**
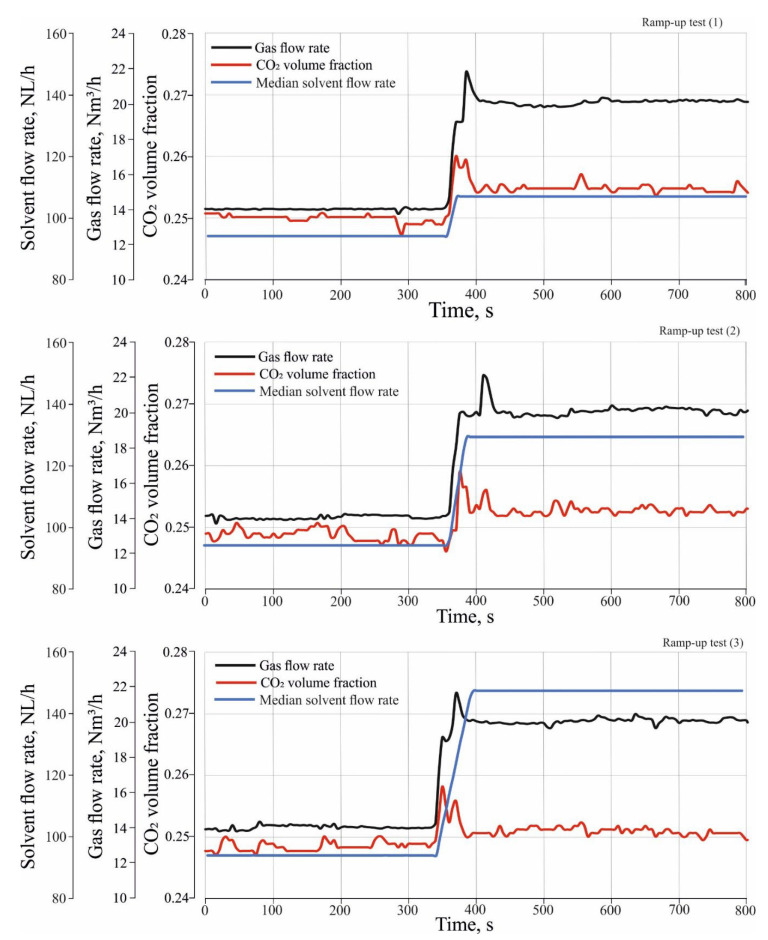
Effect of different load changes in inlet gas and solvent flow rate on outlet CO_2_ volume fraction.

**Figure 6 entropy-24-01318-f006:**
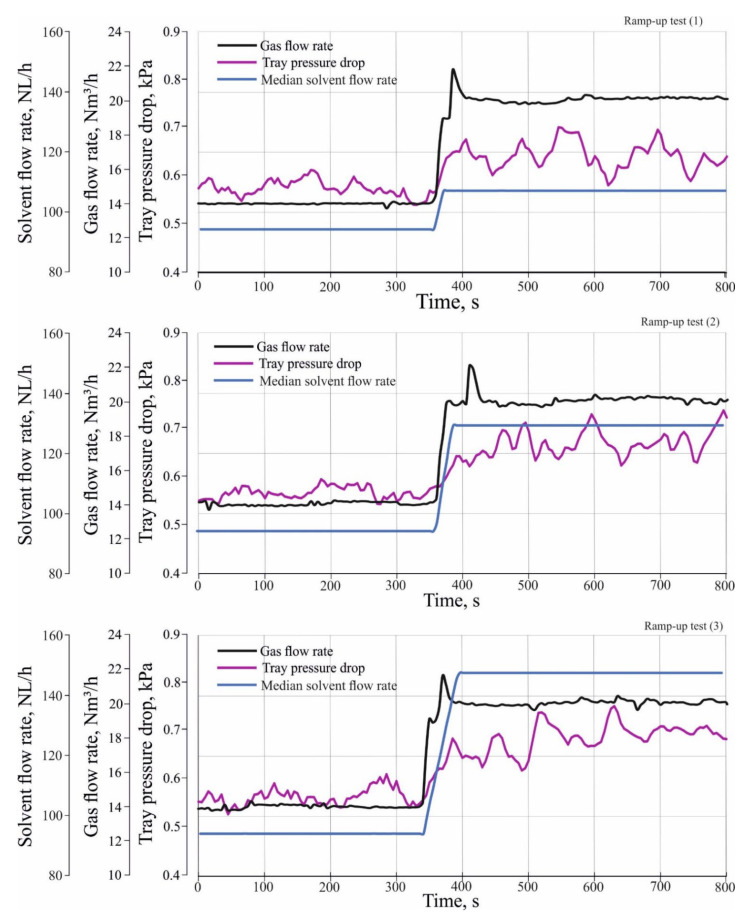
Effect of different load changes in inlet gas and solvent flow rate on total tray pressure drop.

**Figure 7 entropy-24-01318-f007:**
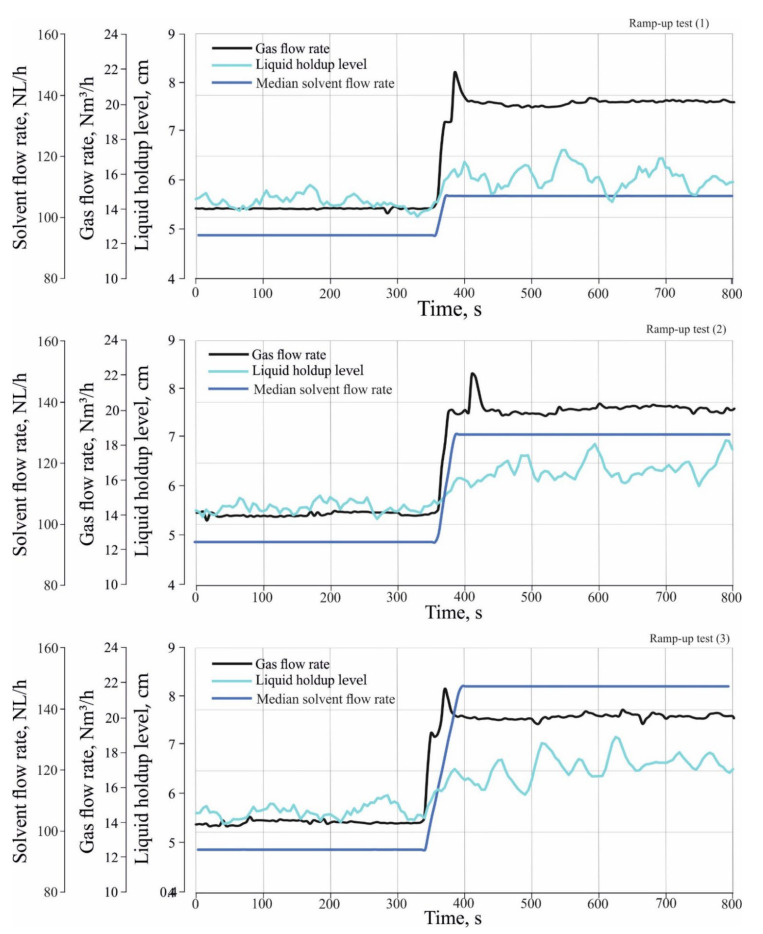
Effect of different load changes in the inlet gas and solvent flow rate on Liquid holdup level.

**Figure 8 entropy-24-01318-f008:**
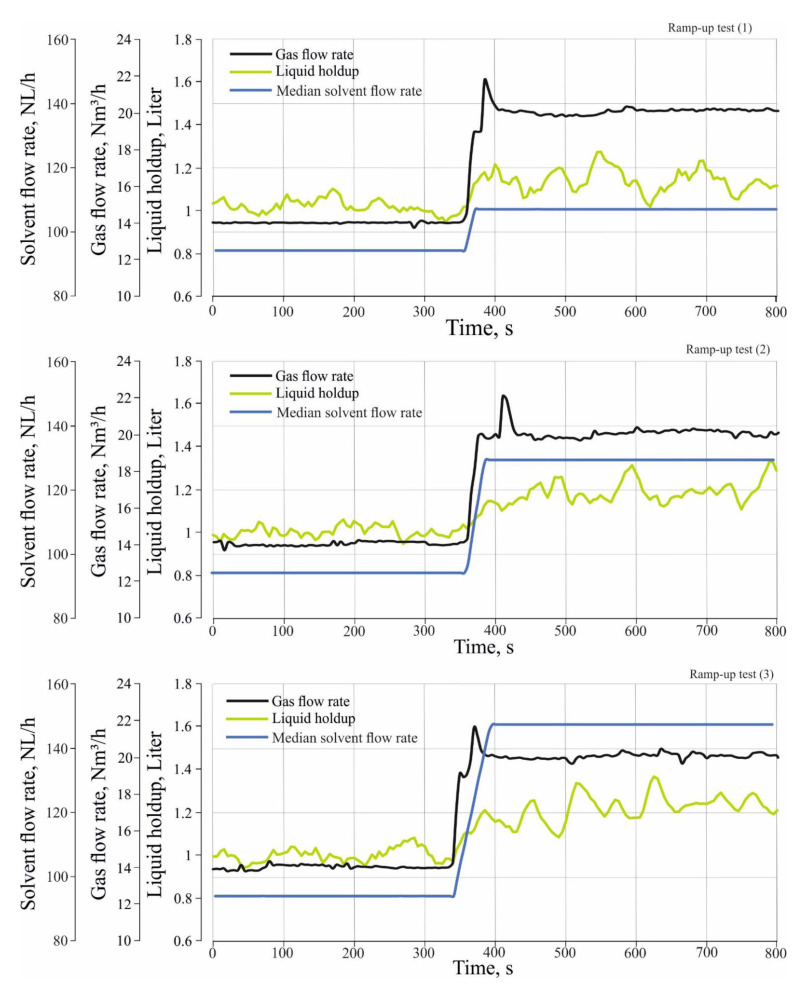
Effect of different load changes in the inlet gas and solvent flow rate on Liquid holdup.

**Figure 9 entropy-24-01318-f009:**
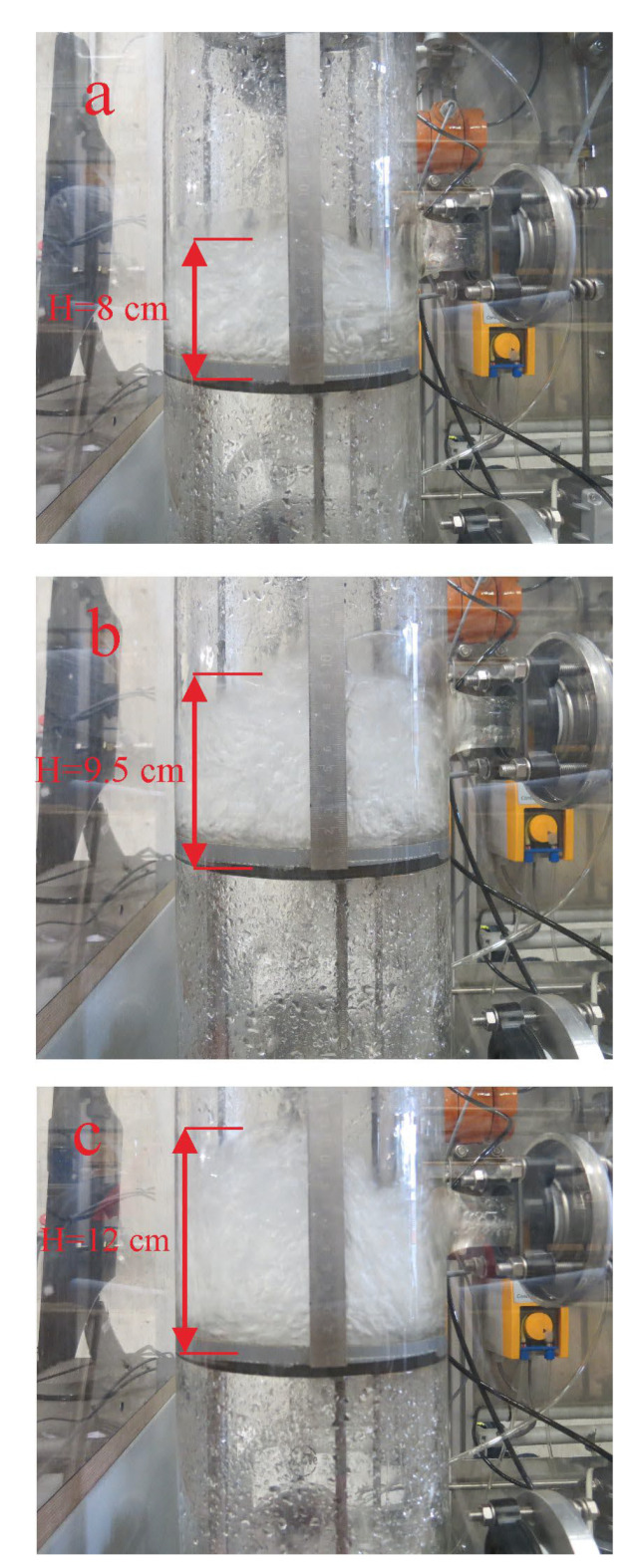
Froth height during the transient state, (**a**) after 1 s, (**b**) after 20 s, and (**c**) after 50 s from running the ramp-up test.

**Figure 10 entropy-24-01318-f010:**
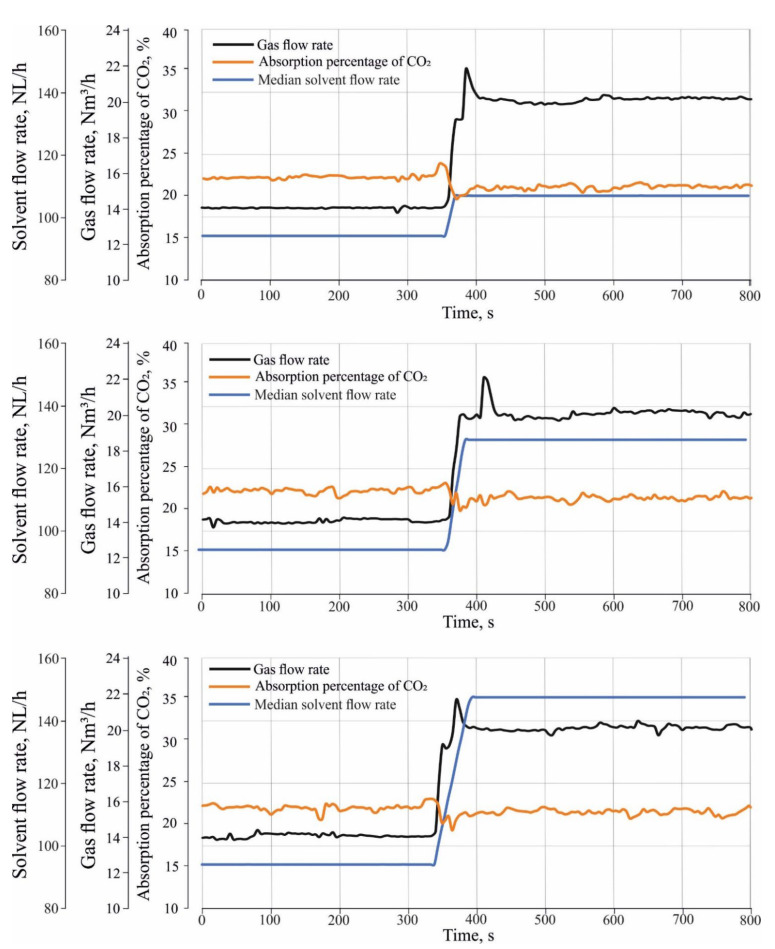
Effect of different load changes in the inlet gas and solvent flow rate on the absorption percentage of CO_2_.

**Table 1 entropy-24-01318-t001:** The measurements of the dry tray pressure drop.

Stationary State	The Dry Tray Pressure Drop [kPa]
Stationary state (1)	0.005
Stationary state (2)	0.007

## Data Availability

Not applicable.

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
