# Peer review of "Experimental Study of the Influence of Different Load Changes in Inlet Gas and Solvent Flow Rate on CO2 Absorption in a Sieve Tray Column"

_entropy, 2022, doi:10.3390/e24091318_

Round 1
Reviewer 1 Report
In this paper, the experiment of absorbing simulated flue gas with MEDA aqueous solution on a test platform for CO2 absorption, and the effect of load change on CO2 absorption was studied. This article has some practical significance, but the innovation is not enough. The overall structure of the paper and the writing of the article also need a lot of improvement, therefore, I recommend rejecting the manuscript.
Major quesitons:
(1) In the section of introduction,the previous researches were list,but lack of organization, summary and analysis, so it is recommended to rewrite.
(2) In the experimental part, the model, manufacturer and country of large-scale instruments and equipment need to be given, such as gas analyzers, reboilers, circulating pumps, etc.
(3) In the discussion part, there is a problem with the definition of absorbed rate. The author's definition is the absorption rate (Nm3/h), not the conventional absorption rate (%).
(4) The number of references is too small, it is suggested to increase.
(5) The language needs to be considerably improved, such as "The Bubbling area is 0.013 m2, the hole diameter on the tray is 2 mm, and the weir height is 15 mm. The tray spacing is 240 mm" on page 4. need proper conjunctions between sentences
Minor questions:
(1) Page 1, there are too many keywords, it is recommended to reduce
(2) Page 4, line 136, correct "0.013 m2" in "Bubbling area".
(3) Page 5, correct the title of Figure 3 "illustrating the geometry of the tray"
(4) Page 6, lines 181-183, correct syntax error, "The pressure control circuit begins controlling the pressure after the gas comes into the absorber, the pressure sensor delivers a signal with the actual value of the pressure to a PID controller "The conjunction is missing between the two sentences.
(5) Page 10, correct "Co2 volume fraction" in Figure 5.
(6) page 11, line 274, correct the obvious grammatical error, "the increase in the superficial velocity of the gas will improve trapping the liquid on a tray and causes the liquid to accumulate on the tray and increasing the total tray pressure drop ."
(7) Page 13, line 300, correct error, "Where The"
(8) Page 13, line 308, correct the typo "Flowing the solvent along the absorber is slower than flowing the gas along the absorber"
(9) Page 13, the units of Liquid holdup level and liquid holdup in Figure 7 are inconsistent with those in the formula.
(10) Page 18, line 368, correct errors "where NCO2 is The absorbed rate", "As It was seen"
Author Response
Dear Sir/Madam,
First, we would like to thank you for reviewing our article and for your comments, which we believe will improve the scientific value of our article. Please see the attachment for updates of the manuscript.
Below you will find the performed revision related to your comments
If you have any questions or comments, please do not hesitate to contact us.
Yours sincerely
Adel Almoslh
Major quesitons:
(1) In the section of introductionthe previous researches were list but lack of organization, summary and analysis, so it is recommended to rewrite.
This has been done. Pages 2-4.
(2) In the experimental part, the model, manufacturer and country of large-scale instruments and equipment need to be given, such as gas analyzers, reboilers, circulating pumps, etc.
This has been done. (Page 6, lines 252-253), (Page 9, lines 273-279).
(3) In the discussion part, there is a problem with the definition of absorbed rate. The author's definition is the absorption rate (Nm3/h), not the conventional absorption rate (%).
This has been done. Page 24, Figure 10.
(4) The number of references is too small, it is suggested to increase.
This has been done. The number of references is increased from 12 to 20, Pages 27-28.
(5) The language needs to be considerably improved, such as "The Bubbling area is 0.013 m2, the hole diameter on the tray is 2 mm, and the weir height is 15 mm. The tray spacing is 240 mm" on page 4. need proper conjunctions between sentences.
This has been done. Page 6, lines 245-246.
Minor questions:
(1) Page 1, there are too many keywords, it is recommended to reduce
This has been done. Page 1, lines 25-26.
(2) Page 4, line 136, correct "0.013 m2" in "Bubbling area".
This has been done. Page 6, line 245.
(3) Page 5, correct the title of Figure 3 "illustrating the geometry of the tray"
This has been done. Page 8, line 264.
(4) Page 6, lines 181-183, correct syntax error, "The pressure control circuit begins controlling the pressure after the gas comes into the absorber, the pressure sensor delivers a signal with the actual value of the pressure to a PID controller "The conjunction is missing between the two sentences.
This has been done. Page 9, lines 298-305.
(5) Page 10, correct "Co2 volume fraction" in Figure 5.
This has been done. Page 14, in Figure 5.
(6) page 11, line 274, correct the obvious grammatical error, "the increase in the superficial velocity of the gas will improve trapping the liquid on a tray and causes the liquid to accumulate on the tray and increasing the total tray pressure drop ."
This has already been done. Page 15, lines 397-400.
(7) Page 13, line 300, correct error, "Where The"
This has been done. Page 17, line 427
(8) Page 13, line 308, correct the typo "Flowing the solvent along the absorber is slower than flowing the gas along the absorber"
This has been done. Page 17, lines 434-435.
(9) Page 13, the units of Liquid holdup level and liquid holdup in Figure 7 are inconsistent with those in the formula.
This has been done. Page 17, lines 418-428.
(10) Page 18, line 368, correct errors "where NCO2 is The absorbed rate", "As It was seen"
This has been done. (Page 22, line 501).

Reviewer 2 Report
In my opinion, the article is interesting and worthy of being considered for publication in this Journal.
However, some details must be adjusted.
Why the authors considered exclusively CO2 and N2 in the gaseous mixture used for experiments?Could the presence (even at low concentrations) of further species modify the results? What about the entity of this potential variation? A brief discussion should be included in the text.
The last sentence of the abstract should be removed. It is not appropriate for the abstract.
The introduction is well structured and the description of the current literature is of quality. However the quantity of studies considered for its production is extremely low and must be widely extended.
Revise supercripts (m2, ecc..)
Revise the caption of figures (format, intial capital letter, final dot, ecc..).
Apart this, the description of the apparatus and the methodology are clear and provide the needed details. The experimental section is of quality.
The language must be revised: there are numerous typos errors diffused along the text.
In conclusion, some revisions are required, but the quality of the article is not under discussion. I opted for minor revision.
Author Response
Dear Sir/Madam,
First, we would like to thank you for reviewing our article and for your comments, which we believe will improve the scientific value of our article. Please see the attachment for updates of the manuscript.
Below you will find the performed revision related to your comments.
If you have any questions or comments, please do not hesitate to contact us.
Yours sincerely
Adel Almoslh
- Why the authors considered exclusively CO2 and N2 in the gaseous mixture used for experiments?Could the presence (even at low concentrations) of further species modify the results? What about the entity of this potential variation? A brief discussion should be included in the text.
This has already been done. Page 10, lines 328—334.
- The last sentence of the abstract should be removed. It is not appropriate for the abstract.
This has already been done. Page 26, lines 582-584.
The introduction is well structured and the description of the current literature is of quality. However the quantity of studies considered for its production is extremely low and must be widely extended.
This has already been done. The number of references is increased from 12 to 20, Pages 27-28.
Revise supercripts (m2, ecc..)
This has been done. Page 6, line 245.
Revise the caption of figures (format, intial capital letter, final dot, ecc..).
This has been done. Figures 1-10.
Apart this, the description of the apparatus and the methodology are clear and provide the needed details. The experimental section is of quality.
The language must be revised: there are numerous typos errors diffused along the text.
This has been done.
In conclusion, some revisions are required, but the quality of the article is not under discussion. I opted for minor revision.

Round 2
Reviewer 1 Report
The author has made revisions to the issues raised in the last review. But there are still some problems, it is recommended to accept after minor revisions:
(1) The language needs to be further refined to meet the publication requirements;
(2) There is a problem with the format of Table 1, which has no lower border.
(3) The conclusions should be simplified, because there are too many conclusions in this version.
Author Response
Dear Sir/Madam,
First, we would like to thank you again for reviewing our article and for your comments, which we think will improve the scientific value of our article. Please see the attachment related to the revised manuscript.
Below you will find the performed revision related to your comments.
If you have any questions or comments, please do not hesitate to contact us.
Yours sincerely
Adel Almoslh
minor revisions:
(1) The language needs to be further refined to meet the publication requirements;
The language has been further refined. The revision was mainly on Pages 12-26.
(2) There is a problem with the format of Table 1, which has no lower border.
A lower border has been added to Table 1.
(3) The conclusions should be simplified, because there are too many conclusions in this version.
The conclusions have been simplified.
